# The Role of Green Infrastructure in Pluvial Flood Management and the Legislation Surrounding It: A Case Study in Bristol, UK

Dudley Saunders  and John Martin *

Sustainable Earth Institute, University of Plymouth, Plymouth PL4 8AA, UK
* Correspondence: j.martin-2@plymouth.ac.uk

**Abstract:** Surface water flooding is an issue which has required an increased level of addressment over the past two decades, with the methods used to combat flood events seeing an evolution. This evolution has been influenced heavily by multi-scale legislations and policies, which has pushed for more holistic methods for pluvial flood management. This review will analyse how Bristol City Council have implemented these multi-scale pieces and what has been put in place to encourage sustainable flood management. This will be done through a purposive review of the literature and an extensive review of legislation and policies on a national, European, and regional scale. The findings of the review were able to show that international legislation and policies are not in place to support sustainable management. UK policies, however, were more supportive, with direct reference and guidance for how to move away from hard engineering solutions. The City of Bristol has embraced the concept of sustainable flood management, with the highest level of support witnessed through the multi-scale review. Overall, the City of Bristol has achieved a good understanding of how to use sustainable drainage, with many systems throughout the city, and schemes to support the use. However, further legislative pieces need to be passed on a national and European scale to encourage and promote the deployment of these systems, so the benefits can be acquired on a large scale.

**Keywords:** sustainable; nature-based solutions; pluvial flood management

## 1. Introduction

Over the past two decades, the UK has seen a huge increase of urbanisation, with cities having the largest increase at 16% growth since 2001 [1]. This urbanisation has forced cities and towns to undergo urban development, in turn leading to a decrease of green spaces, tree coverage, and permeable surfaces, with the UK losing 22,000 hectares of green spaces between 2006 and 2012 [2]. This urbanisation is not expected to fall, with UK city population growth expected to increase by 7.6% from 2015 to 2025 [3].

This urbanised land leads to a reduction in rainfall infiltrating into the ground, creating sheet flows, higher surface runoff, and overall a higher load for water bodies such as rivers, streams, lakes, and ponds [2]. Effects from this can be seen to increase hydrological responses past the natural rate, pushing rainfall events to be increasingly intense and common [4,5]. Climate change predictions also provide evidence of increased intensity and frequency of flood events [6]. This increase of impermeable land and rainfall has been seen to increase flooding, with the frequency and economic losses also rising [6]. Flooding in the UK has increased dramatically since 1884 and is now experiencing the highest frequency recorded costing the economy £1.6 billion in 2015/2016 [7]. Hard engineering for flood defence including drains, sewer systems, and gutters have been commonplace in cities since their inception, as a method of quickly draining rainwater, with the intention of providing enough capacity to prevent pluvial flooding in extreme weather conditions [8]. However, the increasing scale of urbanisation, higher annual rainfall, and climate change have all put pressure onto these systems, requiring more protection and swelling the costs of operating, designing, and maintenance [8].

These factors have made hard engineering solutions to flooding potential open to criticism, and with the current legislations calling for more sustainable methods to be used, Nature-based Solutions (NbS) are growing in popularity. However, the support for their usage in the UK is questionable and could require further legislative and policy-driven support to enhance the effective use of them on a national scale [9,10]. NbS is an umbrella term for a method of conserving and enhancing the environment, via using only sustainable techniques, the benefits also extend to socio-economic growth and public health development [11]. The International Union for Conservation of Nature (IUCN) defines NbS as 'actions to protect, sustainably manage, and restore natural and modified ecosystems that address societal challenges effectively and adaptively, simultaneously benefiting people and nature.' [12]. Organisations such as the IUCN has seen the potential of NbS for assisting in climate change, with issues such as flooding becoming increasingly adverse and NbS enabling a sustainable way of combatting this [11,12].

Over the past two decades, a more holistic, bottom-up approach has been adopted for flood management; this can be traced back to the Millennium Ecosystem Assessment (MEA) report, with the report changing the viewpoint of Ecosystem Services and displaying the advantages to both humans and the environment [13,14]. This transgression from traditional reactive or proactive flood management to a more interactive or holistic approach promises to better equip cities with sustainable methods, cost effectiveness, and variability of flood protection; however, the UK are still yet to fully embrace this method, with the use of NbS relying on independent local flood management groups [14,15]. The movement to more sustainable methods also aligns with the Sustainable Development Goals (SDGs), which were adapted from Rene Passet into an international convention by the UN in 2015, with aims of enhancing principles including social, environmental, and economic wellbeing [16]. NbS systems (e.g., living walls, swales) have the ability to reform the hydrological cycle in urban settings. This in turn can assist in controlling excess rainwater in storm events. This is done by developing ecological elements, for example soil is adept at allowing water to infiltrate and be stored in its pore spaces, and vegetation allows for the hydrological cycle to slow down by infiltrating precipitation with branches and leaves, and intaking water through transpiration [9,16]. Gill et al. (2007) used a surface runoff model to investigate the theorised increase of surface water runoff in urbanised areas surrounding Manchester, UK in the coming century, as an effect of climate change [17]. It was found that rainfall events of 28 mm—which are expected to increase by 55.6% by 2080 in Manchester—will produce 82.2% more surface water runoff; however, by incorporating 10% green cover throughout Manchester, it is expected to reduce this surface water runoff by 4.9%, with tree cover to the same amount decreasing the runoff by a further 5.7% [17]. These simple forms of NbS show how effective even a primitive system can be.

The past decade has seen various studies investigate the most effective NbS for urban pluvial flood risk management [8,14,18,19]. These studies have quantified both the benefits of various methods of NbS and identified the most effective techniques for a range of situations in the urban environment. The studies suggest that using a range of NbS provides the most effective results [9,14,18]. Sustainable Urban Drainage Systems (SuDS) were the most effective variation of NbS for pluvial flood prevention in urban areas, these systems are designed to represent the hydrological cycle and assist in infiltration in impermeable areas [15]. Individual SuDS have been found to reduce the stormwater runoff volumes on impermeable roads by 89–100% throughout Illinois, USA, with bioswale-infiltration trenches having the highest level of effectiveness at 100% [9,18]. However, there are negatives attributed to SuDS, for example, the build-up of sedimentation in the system, which can lead to reduced water capacities and reduced efficiency [19]. This issue is dealt with through dredging, having to be undertaken on average between 7 and 10 years, but in some extreme cases it can be shorter, such as in Belgium where a retention pond was filled with sediment after only four years of use, causing a higher level of funding for this more frequent dredging schedule [20]. Studies have assessed the effectiveness of combined NbS, utilising green roofs and swales in conjunction with other systems to assist

in reducing flow volumes of surface water runoff by 31–42% and peak flows by 5–15%, in the urban setting of Foshan, China [9,21]. This area of research has concluded that NbS are most effective for flood management for 1–15-year flood events, in comparison to 50- or 100-year flood events [9,22]. Due to climate change drastically increasing the amount of smaller scale flooding events that are seen from 1–15 years in comparison to large scale events, this information provides backing for NbS being an effective tool and gives creditability to further implementation in urban environments in the UK [23]. Due to the increasingly adverse and common nature of pluvial flood risk throughout the UK, a critical review of multi-scale legislations and policies regarding NbS for flood management will be undertaken, as well as an evaluation of current systems in place for a case study of choice, in order to provide a reliable analysis of how the threat of pluvial flood management is handled and regarded in UK cities [8,14,15].

This study falls in the planning stage of disaster risk management, enabling an analysis of current legislations surrounding NbS and if these are sufficient.

## 2. Materials and Methods

### 2.1. Case Study Choice

Bristol a city in the west of UK (Figure 1) was selected as a case study due to the city's high urban growth paired with the increasingly large risk of pluvial flooding [24,25]. The city is expecting a population growth from 465,900 to 535,200 (growth of 69,300) over the next 25 years, 50% higher than the average growth of the country for this period [24]. This increase of impermeable land which Bristol is experiencing has played a part in the city being recognised as one of the top 10 flood risk areas to pluvial flooding in the UK, with approximately 22,300 residential properties at risk from pluvial flooding currently [25]. The high pluvial flooding potential paired with the rapid growth of the city makes it a prime candidate for analysing the local governments response to national and international legislation surrounding the use of NbS.

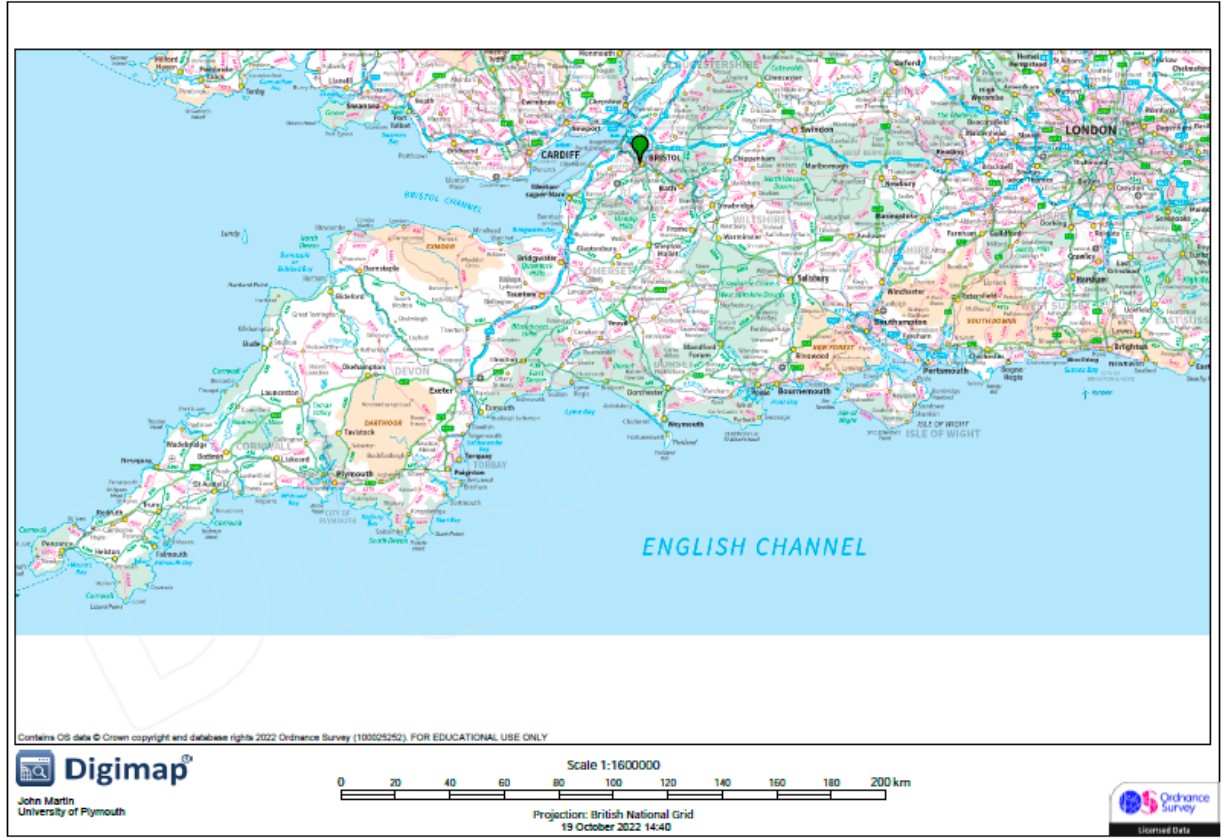

**Figure 1.** Location map of Bristol, green pointer showing location.

*2.2. Method*

This paper relies on a purposive review of grey as well as academic literature, legislation and action plans surrounding pluvial flood management in the UK, by looking at national, European, and regional flood management policies to discover the prevalence, integration and role that NbS play in flood management. This data collection was completed by utilising Cooper's (1988) taxonomy of Literature Review [26]. This multi-scale review will enable an analysis for evaluating the persistence and offering insights which could otherwise be missed, gaining a true insight into the interconnectedness of the policies [27]. As well as this, the paper will investigate specific strategies used in the City of Bristol, assessing the efficiency of these methods with suggestions of improvement being undertaken if required. The review began by conducting extensive research into European and UK national drivers which supported NbS for flood management. The contents of the policies will be analysed based on criteria in order to identify their influence onto regional/local policies, the criteria focused on are as follows:

- Recommends NbS as a technique for flooding; and
- Analysis the development and effectiveness of NbS.

Scopus and WOS were used to search for appropriate literature, with key words "Nature-based Solutions", "Pluvial Flood Management", "Flood Legislation", and "UK" used to recognise articles which would be considered for analysis. Articles would only be considered for use if they were written in English, were published during the time of the research (June 2022), and were written no less than 20 years prior to the research. During the screening of literature, suitability was deemed if original information was presented, a full description of SuDS which have been adopted and not purely a list was given, or an analysis of legislative movements regarding sustainable flood management in the UK at a national, local, or international level.

## 3. Results

The results of this paper have been split into two primary sections, with the first analysing how soft engineering for pluvial flood management has been managed through legislative and policy drivers at multi scale levels, and secondly an in-depth look into how Bristol has implemented these drivers into their own policies, and how effective this has been for deployment of sustainable flood management throughout the city, seen in Figure 3 which shows the SuDS prevalent throughout the city of Bristol as well as the primary habitats. Table 1 shows a breakdown of all relevant multiscale pieces.

**Table 1.** Multi-scale legislation and policy analysis regarding relevant documents surrounding sustainable flood management. Key: The presence of the criteria previously mentioned in these documents are expressed as 'High (+++) 'Medium (++)' and 'Weak (+)'. NM means "not mentioned".

| | Title of Driver | Primary Focus | Evidence for NbS | Recommends Sustainable Management |
|---|---|---|---|---|
| International Legislation | | Improve water quality for a variety of watercourses | NM | + |
| | **European Floods Directive** | Evaluates flood risk and how to manage flooding | NM | + |
| | **Paris Agreement** | Limit adverse effects of climate change | NM | ++ |
| National Legislation | **Flood and Water Management Act** | Provided Infrastructure for flood management authorities to move towards sustainable flood management | ++ | +++ |
| | **Flood Risk Regulations** | A framework for assessing danger regarding flooding and strategies to overcome the risk | NM | NM |
| | **A Green Future 25 Year Environment Plan** | Aims to provide cleaner air and water throughout the UK | +++ | +++ |
| | **Making Space for Water** | Developing flood and coastal risk management | NM | + |
| | **The Water White Paper** | Documents future legislations which may be implemented | ++ | +++ |
| | **National Flood and Coastal Erosion Risk Management Strategy** | Comprehensive strategy of how the UK is aligning with EU Regulations to assist biodiversity | +++ | +++ |
| | **Biodiversity 2022** | To prepare the UK for flood management un-til 2100 | NM | ++ |
| | **Pitt Review** | Recommendations for flood management fol-lowing the serious flu-vial and pluvial flood-ing of 2007 | +++ | +++ |
| Regional | **Bristol Local Flood Risk Management Strategy** | This strategy aims to show measures that will be taken to mini-mise flood risk throughout the city | +++ | +++ |
| | **Bristol Ecological Emergency Action Plan 2021–2025** | This action plan shows how the council plan to prioritise nature with plans going ahead to encourage restoration of wildlife in the city | +++ | +++ |

*3.1. International Legislation and Policy*

European and international policy has been moving towards a more holistic approach to flood management since the MEA assessment of 2005 (MEA, 2005), the multifaceted approach which is able to assist in both flood defence alongside biodiversity and climate change adaptation, has been seen as a pivotal solution to many issues which follow urbanisation [28].

The knowledge surrounding the increasing flood risk present worldwide helped to create two kinds of drivers, either surrounding legislation or policy [29]. The European Floods Directive 2007/60/EC presented regulations surrounding flood management and parameters which were deemed essential to follow [30,31]. It enabled evaluation of the risks caused by flooding into a quantitative value and required data to be collected by individual countries to allow numerical models to further the progress of future flood prediction. This directive was able to disperse and show the information regarding the requirement for more sophisticated and integrated flood management to European governments [31]. Section (13) of the Directive references 'promoting the achievement of environmental objectives'; however, there is no direct mention of using NbS to assist the scheme or of the benefits that it can provide [31]. The international backing for NbS is not excessive, with countries left to determine the involvement they would like regarding sustainable flood management. International conventions such as the Paris Agreement are other examples of the willingness for change on an international scale, the convention aims to limit the global temperature rise to 1.5 °C, and although it does not exclusively mention NbS, the primary objective of the treaty is in alignment with that of sustainable flood management, with the agreement being seen to influence a multitude of areas [32].

*3.2. National Legislation and Policy*

The international and European legislation has been seen to heavily influence how the UK deals with the issue of flooding, with both policy and legislative drivers forming a framework for supporting NbS. One of the primary legislative pieces of governance which recognises NbS has been the Flood and Water Management Act of 2010 [33]. This act aimed to reduce the flood risk associated with extreme weather events, such as rainfall events of >60 mm a day that are expected to increase in relation to climate change [34]. The act provided local authorities, the EA, and water companies more responsibility for flood management.

This act also gave more responsibility to these governing bodies, with Section 7 of the Act requiring the EA to produce a national flood and coastal erosion risk management strategy for England. The document states that a range of resilience actions and techniques will be required for better flood prevention throughout the UK, specifically mentioning NbS as a way to slow flow or store flood waters [33]. The UK government also share the mindset of focusing more on NbS, shown in 2020 by increasing partnership funding to include NbS, pushing its development, increasing the viability of its use, and showing the value of this method.

Throughout the document, NbS are referred to as a technique to protect the environment, enabling management of the flow of water, thus reducing risks of flooding and coastal change [33]. Strategic Objective 1.4 states that the use of NbS will increase between 2020–2030, with NbS having an important role in tackling future flooding and becoming crucial for achieving the government's 25-year Environment Plan [33]. However, this is never quantified through the document, leaving the amount by which it is planned to increase ambiguous. This document does not focus on a nationally managed deployment of NbS, instead placing the responsibility onto local communities as stated during Objective 1.4 'Nature based Solutions provide opportunities for local communities and local groups . . . to become actively involved in how flood resilience is achieved in their local areas'. This strategy has the potential to succeed, with individual areas requiring differing intensity and technical application of NbS, which can be catered for by local authorities which have been given new statutory powers and responsibilities and deemed Lead Local Flood Authorities (LLFAs) [33]. On the other hand, the independence given could lead to negligence and naivety in local flood management, with communities missing the multi-faceted benefits that NbS can provide due to a lack of legislation pushing this agenda.

These LLFAs will work in partnership with organisations such as the EA to develop effective strategies for local flood management. This Local Flood Risk Management Strategy (LFRMS) aims to identify key areas for focus, supplying frameworks for the strategies which will be most successful.

To achieve the objective of reducing flood risk from extreme weather events several measures were stated as part of the Act, relevant measures included;

- 'Measure 1.4.2: From 2021 risk management authorities will work with catchment partnerships, coastal groups, land managers and communities to mainstream the use of nature-based solutions
- Measure 1.4.3: From 2021 risk management authorities will contribute to improving the natural, built and historic environment by investing in projects that manage flood and coastal risks where this is appropriate.
- Measure 1.4.4: From 2021 investments in flood and coastal projects by risk management authorities will help to achieve objectives in river basin management plans and contribute to the government's aim for 75% of waters to be close to their natural state as soon as practicable.
- Measure 1.4.5: From 2021 risk management authorities will work with Natural England and other partners as they develop Local Nature Recovery Strategies that enable new and restored habitats to contribute to flood and coastal resilience.' [33].

This legislation highlights the movement from hard engineering solutions to softer methods, with the advice to utilise both of these strategies for the most effective results, this legislation can clearly be seen to have been adopted due to international advances in the field, including the EU Water Framework Directive of 2000 and the MEA assessment [13,35]. Policies are another area that has seen a dramatic influence of sustainable flood management. One primary national scale policy that is representative of the cumulative changes surrounding sustainable flood management in policies is the Pitt Review; this entails responses to a report published by Sir Michael Pitt following intensive flooding and how to better prepare and respond to events such as these [29,36,37]. Some of the primary points made throughout the review include the need for resolution of issues surrounding who is responsible for the ownership and maintenance of SuDS and a reduction in impermeable surfaces surrounding residential properties [37]. In total, 92 recommendations were made throughout the review, with each of these having responses from both the EA and DEFRA, aiding in the creation of national policies including the Water White Paper, which advises the retrofitting of SuDS into urban environments and influencing Schedule 3 of the Flood and Water Management Act 2010 to encourage the use of SuDS and create a governing body for them [29,36,38].

### 3.3. Regional Legislation and Policy

The national and international response to sustainable flood management has started to show impacts on a regional scale, despite the UK government's choice to give more responsibility to LLFAs instead of implementing national scale legislations regarding sustainable management. The encouragement of local level authorities to manage flooding has produced a multitude of Action Plans, policies, and development which can be seen to vary hugely between districts or counties. Bristol's flood risk is primarily managed by the City of Bristol Council, which were chosen as the LLFA of the city after the Flood and Water Management Act of 2010; they were therefore given the responsibility to incorporate a LFRMS for the city. This scheme was created in 2014, the purpose of this strategy is to:

- 'Provide an overview of flood risk in Bristol;
- Explain the role of organisations involved in flood risk management;
- Set out the objectives for managing local flood risk;
- Put in place measures to achieve the objectives;
- Produce an action plan that explains how and when the measures are to be implemented;
- Examine the costs and benefits of delivering the measures; and

- Demonstrate how the strategy contributes to the achievement of wider environmental objectives' [25].

Approximately 300 houses have been protected from flooding since its creation, with this number due to rise significantly in the coming years due to issues including climate change and urbanisation [25]. The LFRMS contains a risk assessment of the potential of flooding from a variety of ways, including surface water, tidal, fluvial, groundwater, and sewers; however the document states the risk from surface water flooding is the most problematic when modelling future scenarios [25].

Studies analysing the risk of surface water flooding have found that over 22,000 residential properties in Bristol are at risk of surface water flooding, gaining Bristol a position as one of the UK's top 10 Flood Risk Areas for surface water flooding [25]. Modelling of climate change into the effects of surface water flooding were also carried out for this document, with studies concluding that Bristol faces a significantly increased risk of flooding from surface water in the coming years due to higher extreme rainfall events which the city is not currently equipped to tackle [25].

The first mention of sustainable flood management in this document is found in the guiding principles of the LLFA, where sustainability is stated as one of the six top principles which are followed for every decision made in the LFRMS [25]. The guidance states 'Wherever possible, solutions to flooding problems should work with natural processes and aim to enhance the environment' [25]. The national pieces of legislation and policy surrounding sustainable flood management can be seen shadowed in this LFRMS, with strategy's including 'preventing installation of impermeable surfaces without using SuDS to assist in runoff', having a clear influence from the Pitt Review and the subsequent reforms that DEFRA committed themselves to [25,36,37].

Another piece of policy for Bristol which refers to sustainable flood management is the Ecological Emergency Action Plan 2021–2025, this action plan aims to support the cities Ecological Emergency Strategy, this policy tries to integrate the best ecological practise into each area of what the council does [39]. Various themes span the action plan, with holistic flood approaches under the titles 'Space for Nature' and 'Water Quality'. E.7 states that further support for SuDS will be undertaken through initiatives including 'Living Roofs', with G.2 advancing on these promises with more guarantees surrounding the plan to embed NbS into LFRMS, these pieces show the effect that multi-scale legislation and policies have had onto the decision making of City of Bristol Council [39].

### 3.4. Current NbS for Pluvial Flood Management in Bristol

Despite the research into utilising multiple NbS for effective management, many SuDS that have been researched for Bristol focus on a singular approach (see Table A1). The table shows that nearly half of SuDS found in the City of Bristol (24 singular SuDS or 46%) are functioning with a singular system, not gaining the extra efficiency which would be provided with a more complex, multiple system approach [9,17]. A total of 52 SuDS are either completed or under construction in the city, with a total of 28 SuDS (53.84%) being within 200 m of a woodland or green space [40].

Figure 3 shows a habitat map of the City of Bristol, adapted from Digimap and City of Bristol Flood Management [40,41]. From this map, four primary habitats have been determined, with urban, suburban, improved grassland, and broadleaved woodland, the majority of the city can be determined as suburban, with a residential population of 465,900 [24]. The map of Bristol also shows the high-risk areas of surface water flooding in the city provided by the Bristol Flood Risk Management team, these areas have a lack of SuDS surrounding them, with only 3 of the 56 having plans for onsite SuDS [40]. Figure 2 shows a breakdown of the varying methods of SuDS used in the city, with the most common choice being permeable paving and soakaways, covering 49.8% of all SuDS used in the city. The least common SuDS included water gardens, green roofs, ponds, and tree pits covering a combined 7.4% of the total SuDS used in the city [40]. The primary location of the SuDS

used in the city are in central Bristol with 24% of the SuDS found here. This area has the least percentage of green space or woodland across the entirety of the city with only 5.8% of ground being permeable in this area [40].

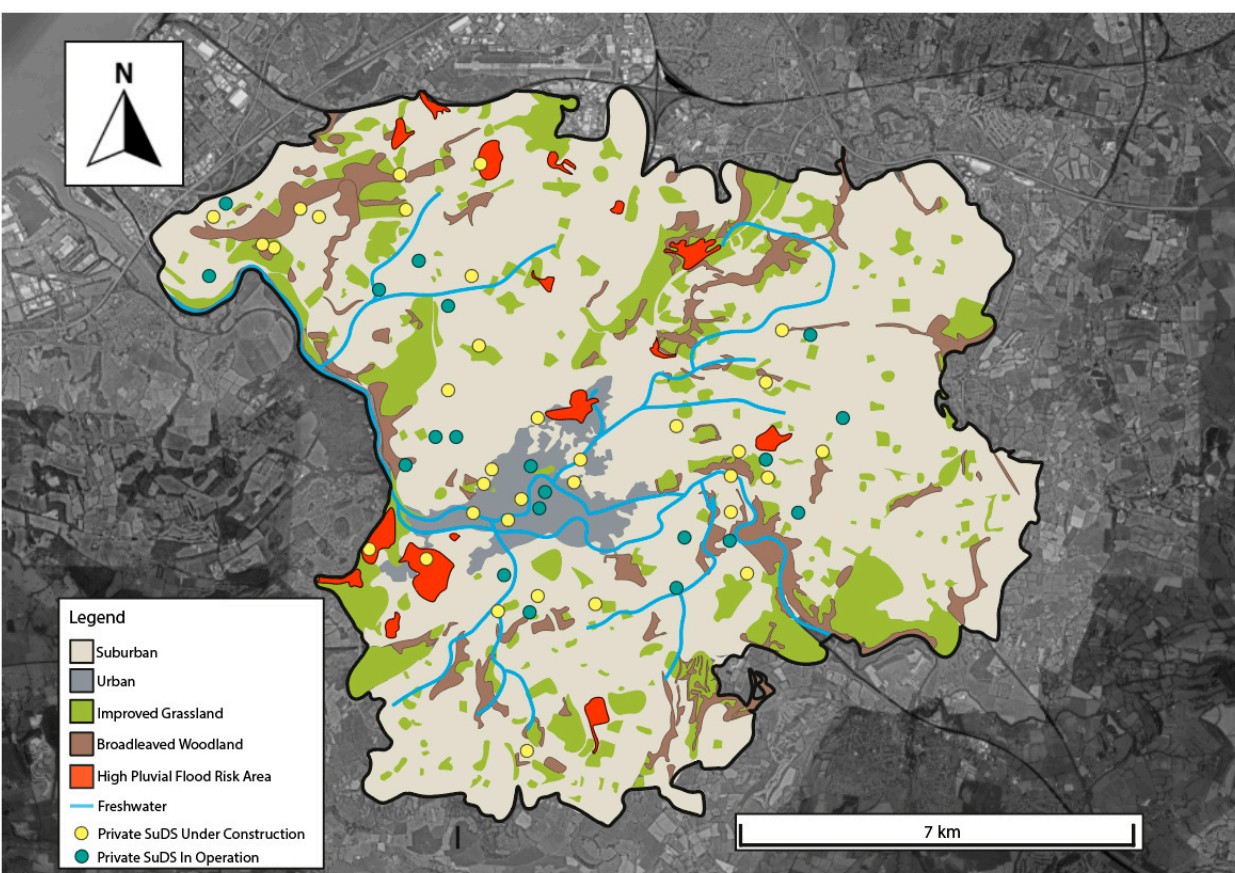

**Figure 2.** Chart showing the varying NbS used in the City of Bristol [25].

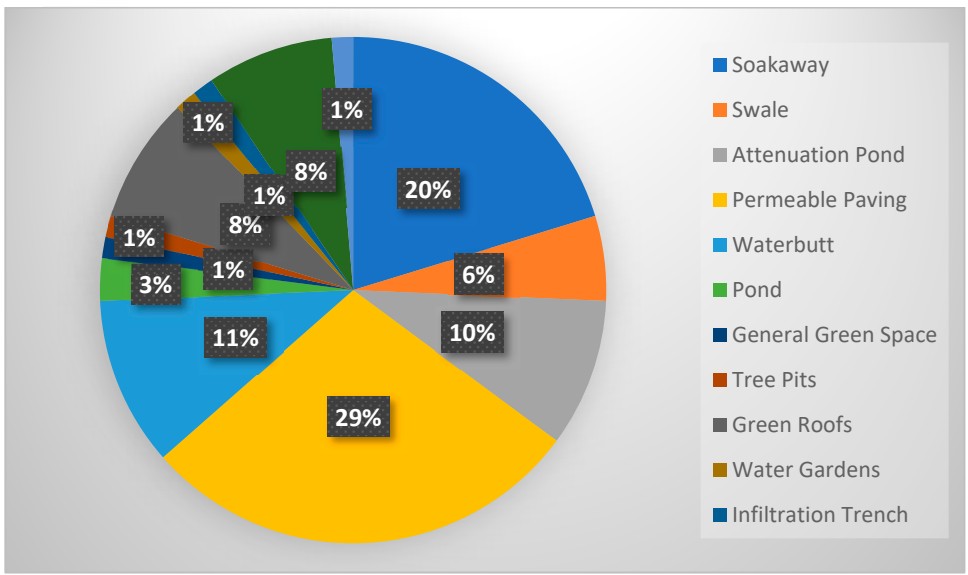

**Figure 3.** Land use map of Bristol, showing SuDS both in use and under construction, using data from Bristol Flood Team [40].

## 4. Discussion

The focus of urban flood management since the 19th century has been primarily centred around the movement of surface water to a more suitable location; these locations mainly included nearby watercourses [15]. This practice is completed by using a network of underground pipes, connected by drainage to the surface, with the pipes having supposed capacity to deal with surface water runoff as well as sewage treatment [15]. This system has until now been largely effective, protecting city dwellers from excessive flooding and assisting farmers in cultivating produce or livestock up to the coast [42]. However, due to climate change and urbanization, modelling has shown these systems are no longer reliable, with multiple large flood events occurring in the past decade:

- Storm Desmond during 2015 caused a new daily rainfall record with 341.1 mm and flooded approximately 5000 homes in Lancashire and Cumbria. The effects of the storm were experienced throughout northern England and southern Scotland [34].
- Flood events throughout the winter of 2019–2020, this extended flood event started in November 2019, primarily affecting Yorkshire and the Midlands before the second stage of flooding began caused by Storms Ciara and Dennis in February of 2020 (Sefton, et al., 2021). The extreme rainfall saw the wettest February since records began in 1766, averaging 169.9 millimetres across the country for the month [43].
- During December 2016, Cumbria experienced extreme rainfall events throughout the month, with 405 mm of rainfall in 48 h flooding 16,000 houses [30,44].

Flood management has been seen to divert from hard engineering to a softer approach. This includes Sustainable Drainage Systems (SuDS) which are one of the most common forms of soft engineering for flood defence [16]. SuDS have the ability to assist in the quality and flow of water sources and have become essential in planning applications in the UK, with benefits spanning biodiversity, socio-economic uses, and flood defences covering all aspects of the sustainability principles [45]. The implementation of SuDS throughout the UK has been primarily focused on a singular system, with one type of SuDS used for a selected area [15]. However, it has become known that utilising an approach of multiple SuDS is more effective, with treatment trains providing the highest level of effectiveness for both flood prevention and water quality benefits [9,18,46].

Based on the literature reviewed, international legislation, and policy regarding NbS, there is not extensive international backing for this concept, the primary pieces of which can be interpreted as being in support are the European Flood Directive 2007/60/EC and the Millennium Ecosystem Assessment (MEA) 2005 [13,31]. However, these pieces all fail to exclusively mention NbS, with the European Flood Directive calling for environmental objectives to be prioritised further in all projects and the MEA assessment providing similar advice [13,31]. The pieces do not focus enough on changes that are occurring, including effects of urbanisation and climate change. This leads to a lack of foresight in the documents which require updates or subsequent pieces to provide guidance for sustainable flood management. With climate change increasing global flood events and urbanisation removing permeable land, now is a priority time to enforce international pieces regarding sustainable flood management. One convention which has seen similar objectives to what is required on a larger scale is the Convention on the Protection of the Rhine. the aim of this convention is to increase biodiversity in the river, reduce pollution, and promote sustainability [47]. The convention promotes restoring natural functions of the river. Although this differs to the focus of this study which analyses pluvial rather than fluvial functions, it still encourages returning to a more natural system, a primary attribute of NbS [47]. This employment of the convention was successful in controlling and enhancing the water quality of the Rhine, as well as restoring the natural flow of the river, reducing flooding events along the five countries that the river travels through [47,48]. Employing a convention similar to this is recommended on a national scale for the UK, promoting sustainable management of pluvial flooding and enforcing stricter policies in relation to this.

Despite the lack of an independent convention, the popularity and knowledge surrounding the importance of sustainable flood management is still developing. The UK's policies and legislations surrounding flood management has changed drastically over the past 20-years, evolving from a hard engineering viewpoint to the realisation of requiring sustainability and the multi-faceted benefits that this can provide, undoubtedly influenced by the international pieces mentioned previously, including the MEA and the Paris Agreement [13,32]. One of the primary documents which changed the way flood management is conducted in the UK was the 'Green Future: 25 Year Environment Plan' [49]. This document shows strong support for sustainable flood management, stating that nature will be used to directly assist with flood management, the piece also acknowledged climate change, with mention of the predicted increase in flooding that can be expected due to this [49]. Although this policy piece alongside other reports have stated the importance and benefits of NbS, there is little reflected in national legislation, with only one legislative piece providing support for sustainable flood management in the Flood and Water Management Act of 2010 [33].

This act was instrumental in the way that pluvial flood management is handled throughout the UK currently, showing high levels of support for NbS, with specific sections titled 'Sustainable development' where a requirement is made for flood authorities to utilise sustainability when undertaking flood management [33]. Alongside this legislation, guidance was issued for how to achieve sustainability, with advice provided promoting SuDS and NbS as ways to sustainably manage flood risk [50]. These were huge steps for how flood management was perceived at a national level, and with these changes came a new perception of independence for pluvial flood management, with the act providing each local authority with an LLFAs [33,50]. This change from a nationally led approach to more independence has meant that guidance surrounding the implementation of sustainable techniques has had the potential to be overlooked by LLFAs; with the supporting pieces being purely guidance, it is difficult for the UK government to enforce specific management styles for areas [14,33,50].

One area which has seen issues with sustainable flood development in the UK has been Newcastle. This was highlighted in O'Donnell's case study, which found certain issues with the decentralised systems incorporated in the area [51,52]. These problems came partially from funding opportunities; maintenance costs from NbS, especially SuDS can be high, this was seen to leave investors disincentivised to provide funding and created a barrier in implementing SuDS through the city [51,52]. This study shows the issues with the independence given for pluvial flood management. If this was instead a centralised system, it would reduce problems with knowledge surrounding the use of NbS and clarify the responsibilities to individual parties, as these were seen as the primary concerns in Newcastle [52]. This overall would drive the use of sustainable management up as it would be enforced on a national scale, rather than left as an independent choice.

However, Bristol has been seen to encourage NbS, with the LLFA of the area implementing various policy measures to assist in creating a more sustainable management style [25]. These policy pieces have shown the emphasis on utilising NbS for flood management and demonstrated the knowledge that the City of Bristol Council have understood the multifaceted benefits they can provide, presenting more support than the official national and international legislative pieces [40]. Both pieces of relevant regional flood policy for Bristol received the highest scores possible on the multi-scale review, surpassing that of the national level. This display of recognition and support is very important for NbS and flood management; however, there are points of improvement which can be made with the deployment seen in Bristol. The high pluvial flood risk areas identified through modelling by City of Bristol Council are at various locations across Bristol, occurring primarily in the suburban areas in the central belt of the city. In total, only 20% of these identified areas have or are within 100 m of a significant SuDS system, demonstrating a lack of support for these high-risk areas [40].

The high surface water risk areas which are scattered across Bristol are primarily habitats such as those seen in Figure 4, residential areas with little to no green space, including lack of a vegetated garden and community green areas, with 68.75% of the high-risk areas fitting into this category [40].

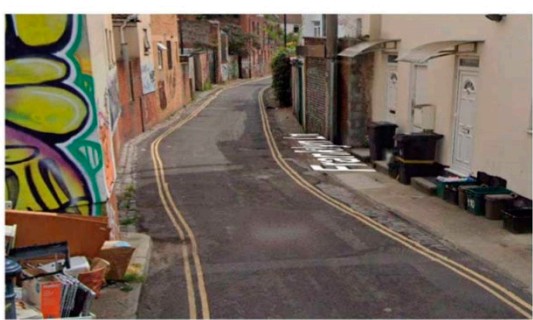 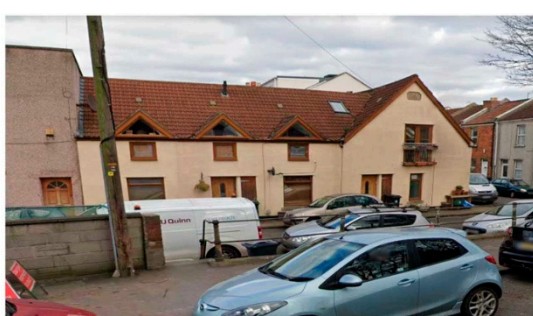

**Figure 4.** Showing areas of high surface water flood risk identified by Bristol City Council (Bristol City Council, 2022).

A potential reasoning for the lack of SuDS in these areas could relate to the public nature of the space, as stated in the SuDS manual; this can have the potential of limiting implementation of SuDS, but more opportunistic thinking can be utilised, which in turn can create many opportunities in these vulnerable areas [9,53]. The most effective SuDS which could be used in these high-risk residential areas include green roofs, living walls, and permeable paving. This is due to the dense urban nature of these spaces, with all these options removing no space from the residents of the areas [29,46,53]. Applying changes such as these would enable a reduction of risk to these areas, at a low costing in comparison to traditional hard flood management and reparations for potential future flooding [51].

Projects setting up encouragement of SuDS are present in Bristol, such as the SuDS in schools partnership with Wessex Water. This scheme provides co-benefits for each party, with an education team conducting sessions with the students, and SuDS being set up over the schools private premises [54]. Examples of this scheme are present in Bristol; however, the lack of power that water companies have in enforcing deployment of these schemes leaves individuals who are uneducated in this field primarily being the heads of schools making decisions which they could be uneducated on [40]. Currently, there are three of these schemes present throughout Bristol. If the deployment of this became mandatory for all education centres, this figure would increase to 136 locations, drastically improving natural flood management in some of the most urbanised areas of the city, as well as providing an education to staff and students about the reality of urban flooding [24], providing environmental and socio-economic benefits to approximately 90,000 people in this age demographic [24]. This lack of legislation and guided approach which has been adopted through the Flood and Water Management Act of 2010 is not sufficient to supply the tools necessary to reduce current and future flood risk for rapidly urbanising areas such as Bristol, with climate change increasing the mean annual flood events in the UK from 12 to 18 over the past century further implementation is strongly required [6].

## 5. Limitations

The limitations surrounding this study are primarily due to its desk-based nature. One major limitation was the inability to complete tests for the effectiveness of sustainable flood management measures. Field investigations were outside of the research scope, but this meant that the research findings were unable to be ratified by testing. This limitation means that the methods of data collection including scientific papers and studies have been relied upon for results and suggestions for improvement. However, relying on scientific data of other areas could be unreliable due to local circumstances

playing a strong role in flood management. To increase the validity of the results a follow up study to this paper is recommended. This follow up would test the theories and forms of NbS that have been suggested for the City of Bristol, assessing the effectiveness of current and proposed NbS for surface water flooding. The second limitation of this study is in regard to contact with the City of Bristol Council's Flood Management Team: due to poor communication lines contact was not possible with the team. This led to all reviews of local policy surrounding NbS being conducted by analysing action plans, local flood management plans, and any other documents of this nature which mentioned NbS. Due to this choice of reviewing policy without direct consultation with City of Bristol Council, the results potentially do not include all NbS measures that the council have introduced and could lead to conclusions surrounding the implantations that are not correct. However, due to the lack of communication, this was a step which had to be taken.

## 6. Conclusions

In conclusion, there has been a large amount of development for sustainable flood management over the past two decades, with the UK releasing multiple policies aimed at enhancing the efficiency and use of NbS [9,49]. Unfortunately, due to the advisory nature of the policies, choices regarding flood management have been left to local management groups, with some areas lagging in the development of sustainable methods, and others embracing the choice of new methods. To address these issues, this paper advises legislation on a national scale, with specific pieces aimed at sustainably managing pluvial flood risk in urban zones, thus assisting in the continued deployment of NbS and pushing areas to be more open to this management style. Overall, adopting this national scale approach would push for a more stable style, with a centralised method allowing easier comparison between areas and utilising the knowledge of government teams such as the EA to provide the best advice. As the area's LLFA, City of Bristol Council have been seen to embrace sustainable management, with the LFRMS showing extremely strong support of these methods. This has transpired into NbS being present throughout the city, for private developments and public spaces [40]. However, as identified by the area's LLFA, there are specific areas of the city which are under high risk from pluvial flooding. These areas show little NbS over all locations and demonstrate a potential lack of foresight in the deployment pluvial flood management [25]. By focusing on retrofitting of NbS, high risk areas such as these would be protected, as stated by Ciria in the manual for SuDS [53].

On an international scale, this evolution of policy has not been witnessed, with no direct mentions of NbS or sustainable pluvial flood management in any international piece [31,35]. An international convention focused on the subject of sustainable flood management through the use of sustainable drainage and the use of ecosystem services would be pivotal for reducing the flood risk throughout urbanising areas and developing a more biodiverse, socio-economically valued area.

**Author Contributions:** Conceptualisation, D.S. and J.M.; methodology, D.S. and J.M.; writing—original draft preparation, D.S. and J.M.; writing—review and editing, D.S. and J.M.; supervision, J.M. All authors have read and agreed to the published version of the manuscript.

**Funding:** This research received no external funding.

**Institutional Review Board Statement:** Not applicable.

**Informed Consent Statement:** Not applicable.

**Data Availability Statement:** Please refer to the following websites for flood data from Bristol City Council at https://maps.bristol.gov.uk/bfrm/ (accessed on 24 September 2022). Please refer to the following websites for habitat data at https://digimap.edina.ac.uk/roam/map/environment (accessed on 24 September 2022).

**Acknowledgments:** I would like to acknowledge the assistance of my partner, Yasmine Garland, for her continued patience.

**Conflicts of Interest:** The authors declare no conflict of interest.

## Appendix A

**Table A1.** Private SuDS register for the City of Bristol, showing singular systems and various types used.

| Private SuDS Register | Type Used | Singular |
| --- | --- | --- |
| 14/00307/F | Soakaway | X |
| 15/01988/F | Swale | X |
| 13/04196/F | Swale and attenuation | |
| 15/02309/F | Permeable Paving and Soakaways | |
| 15/01317/F | Waterbutt | X |
| 13/02550/F | Permeable Paving | X |
| 15/05511/M | Swales, Permeable Paving | |
| 15/01157/F | Unspecified | |
| 12/03634/F | Permeable Paving, Soakaway | |
| 16/02925/M | Waterbutt, Pond, Green Space | |
| 14/01368/F | Tree Pits, Green Roofs, Water Gardens | |
| 14/05459/F | Permeable Paving | X |
| 14/01111/F | Permeable Paving | X |
| 13/05354/F | Green Roof | X |
| 14/01069/F | Permeable Paving | X |
| 15/04154/P | Soakaway, Permeable Paving | |
| 14/02916/F | Soakaway, Permeable Paving | |
| 15/00687/F | Soakaway | X |
| 14/02419/F | Soakaway | X |
| 11/01002/F | Soakaway, Permeable Paving | |
| 15/03273/F | Soakaway | X |
| 13/04417/F | Soakaway | X |
| 10/04750/F | Soakaway | X |
| 14/03076/F | Unspecified | |
| 14/02829/FB | Permeable Paving | X |
| 16/00336/FB | Permeable Paving, Tree Pits | |
| 16/01363/F | Waterbutt | X |
| 14/04161/F | Permeable Paving | X |
| 15/06483/F | Attenuation Tank | X |
| 13/05273/F | Green Roof | X |
| 15/02984/F | Green Roof, Soakaways, Infiltration Trench | |
| 15/02496/F | Green roofs, Attenuation Tank, and Waterbutt | |
| 13/04514/COU | Permeable Paving, | X |
| 13/04567/F | Permeable Paving, Soakaway | |
| 13/05209/F | Green Roof | X |
| 15/01681/F | Pond | X |
| 13/04405/F | Soakaway, Waterbutts | |
| 15/00488/F | Attenuation Tank, Permeable Paving | |
| 14/01347/F | Attenuation Tank, Permeable Paving | |
| 14/06298/X | Attenuation Tank | X |
| 13/04796/F | Permeable Paving | X |
| 16/00537/F | Attenuation Tank | X |
| 15/03408/F | Bioretention Area, Permeable Paving | |
| 15/03407/F | Bioretention Area, Permeable Paving | |
| 15/03308/F | Waterbutts and Bioretention Planters | |
| 15/03305/F | Unspecified | |
| 15/03403/F | Bioretention Area, Permeable Paving | |
| 14/05576/F | Unspecified | |
| 13/04165/F | Soakaway, Green Spaces | |
| 14/03909/FB | Dry Swale, Hydrobrake | |

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
