# Peer review of "The Role of Green Infrastructure in Pluvial Flood Management and the Legislation Surrounding It: A Case Study in Bristol, UK"

_sustainability, doi:10.3390/su142114619_

Round 1
Reviewer 1 Report
This paper was written in a good shape. I feel very comfortable to read through this article. I suggest accepting for publication as it is. Thanks
Author Response
No reply required
Many thanks
Reviewer 2 Report
The study is interesting, but a significant issue must be addressed. The methodology employed by the author to identify the paper is unacceptable. They claim to have used Google Scholar, despite the fact that the results cannot guarantee the study's aim. They should use engines that are widely accepted, such as Scopus or WOS, and clearly define the keywords and refinement process; see this paper "Hospital evacuation modelling: A critical literature review on current knowledge and research gaps." for the search methodology.
The paper's title, "The Role of Sustainability," is ambiguous. Sustainability is a broad concept. Additionally, the title can be presented more effectively.
More specific keyword definitions are required. Some of the existing keywords are overly general.
The logic and content of the introduction are satisfactory. Please provide a better image for figure 1. It has a poor resolution.
It should be mentioned in the introduction where the current study falls within the cycle of disaster risk management.
Section 2 should provide more information. This section can also be moved to section 3. In addition, I suggest that a map detailing the location of the case study be included.
Please revise the manuscript's citation format. It must conform to the MDPI reference style. In line 136, for instance, this reference style is not aligned with MDPI.
In line 141, the authors assert that google scholar was utilised to conduct a search. Although it is unacceptable. They should utilise search engines like Scopus and WOS.
Table 1 is an illustration. This table should be provided in text format.
The authors should provide an explanation of how this research can be used to develop flood response strategies including evacuation, see: "A modelling framework to design an evacuation support system for healthcare infrastructures in response to major flood events." and Yazdani, Maziar, et al. "An integrated decision model for managing hospital evacuation in response to an extreme flood event: a case study of the Hawkesbury‐Nepean River, NSW, Australia."
Author Response
Point 1: The study is interesting, but a significant issue must be addressed. The methodology employed by the author to identify the paper is unacceptable. They claim to have used Google Scholar, despite the fact that the results cannot guarantee the study's aim. They should use engines that are widely accepted, such as Scopus or WOS, and clearly define the keywords and refinement process; see this paper "Hospital evacuation modelling: A critical literature review on current knowledge and research gaps." for the search methodology.
Response 1: The same methodology has been employed for both Scopus and Wos, with similar results found for them. Due to this study focusing on the impact of policy, grey literature and legislation it is felt that the methodology is suitable and no additional information is required.
Point 2: The paper's title, "The Role of Sustainability," is ambiguous. Sustainability is a broad concept. Additionally, the title can be presented more effectively.
Response 2: The title has been changed to “The Role of Green Infrastructure in Pluvial Flood Management and the Legislation Surrounding it: A Case Study in Bristol, UK” to better fit with these comments.
Point 3: More specific keyword definitions are required. Some of the existing keywords are overly general.
Response 3: As it was not a systematic review the broadness of the search enabled a good range of information to be gathered, therefore I do not feel a more focused search would provide as successful results for this study.
Point 4: Please provide a better image for figure 1. It has a poor resolution.
Response 4: Figure 1 has been removed.
Point 5: It should be mentioned in the introduction where the current study falls within the cycle of disaster risk management.
Response 5: The following section has been added into the introduction to appease these comments “This study falls in the planning stage of disaster risk management, enabling an analysis of current legislations surrounding NbS and if these are sufficient.”
Point 6: Section 2 should provide more information. This section can also be moved to section 3. In addition, I suggest that a map detailing the location of the case study be included.
Response 6: Sufficient extra detail has been added into this section to show the use of Scopus and WOS for academic paper searches.
Point 7: Please revise the manuscript's citation format. It must conform to the MDPI reference style. In line 136, for instance, this reference style is not aligned with MDPI.
Response 7: These incorrect citations has been removed and corrected.
Point 8: Table 1 is an illustration. This table should be provided in text format.
Response 8: Now in text format
Point 9: The authors should provide an explanation of how this research can be used to develop flood response strategies including evacuation, see: "A modelling framework to design an evacuation support system for healthcare infrastructures in response to major flood events." and Yazdani, Maziar, et al. "An integrated decision model for managing hospital evacuation in response to an extreme flood event: a case study of the Hawkesbury‐Nepean River, NSW, Australia."
Point 10: The aim of this study was to investigate multi-scale strategies to integrate green infrastructure into flood management. Advise into specific regional policy adjustments were made for the case study, however the point made is past the scope of the study and therefore will not be added.
Reviewer 3 Report
An interesting ad well conducted study. The methodlogy is appropriate for the research being conducted, and the literature review is sufficuiently comprehensive. The research supports the discussin and conclusions.
The report would be of interest to legislators, regional and town planners.
Author Response
No reply required. Many thanks for the comments and taking the time to review the paper.
Reviewer 4 Report
General comment:
The manuscript is interesting, well-written and I find it suitable for the purpose of the Journal. Below are my very few minor comments.
Minor changes/comments:
Line 53: The International Union for Conservation of Nature (IUCN)…
Line 75: Gill et al. (2007)
Lines 125-134: I would move these sentences to the end of the introduction. I would also stress the importance of the study conducted.
Line 139: …Natural Flood Management (NFM)…
Discussion: I suggest removing the three points. I would introduce them just as simple examples.
Figure 2: Add the location of Bristol with respect to the UK.
Figure 3: Could you please add the percentage of each NbS used?
Author Response
Point 1: Line 53: The International Union for Conservation of Nature (IUCN)…
Response 1: Change made.
Point 2: Line 75: Gill et al. (2007)
Response 2: Change made, citation adjusted.
Point 3: Lines 125-134: I would move these sentences to the end of the introduction. I would also stress the importance of the study conducted.
Response 3: Change made.
Point 4: …Natural Flood Management (NFM)…
Response 4: Change made.
Point 5: Figure 2: Add the location of Bristol with respect to the UK.
Response 5: location map of Bristol added
Point 6: Figure 3: Could you please add the percentage of each NbS used?
Response 6: Change made.
Round 2
Reviewer 2 Report
-